# Performance Bounds for Active Binary Testing with Information Maximization

## Abstract

In many applications like experimental design, group testing, medical diagnosis, and active testing, the state of a random variable $Y$ is revealed by successively observing the outcomes of binary tests about $Y$, where new tests are selected adaptively based on the history of outcomes observed so far. If the number of states of $Y$ is finite, the process ends when $Y$ can be predicted with a desired level of confidence or all available tests have been used. Finding the strategy that minimizes the expected number of tests needed to predict $Y$ is virtually impossible in most real applications due to high dimensions. Therefore, the commonly used strategy is the greedy heuristic of information maximization that selects tests sequentially in order of information gain. However, this can be far from optimal for certain families of tests. In this paper, we argue that in most practical settings, for a given set of tests, there exists a $0 \ll \delta \ll \frac{1}{2}$, such that in every iteration of the greedy strategy, the selected binary test will have conditional probability of being 'true', given the history, within $\delta$ units of one-half. Under this assumption, we first study the performance of the greedy strategy for the simpler case of oracle tests, that is, when all tests are functions of $Y$, and obtain tighter bounds than previously reported in literature. Subsequently, under the same assumption, we extend our analysis to incorporate noise in the test outcomes. In particular, we assume the outcomes are corrupted through a binary symmetric channel and obtain bounds on the expected number of tests needed to make accurate predictions.

## 1 Introduction

Many applications of machine learning in science and engineering can be posed as an *active testing* problem of sequentially carrying out tests to predict a target variable $Y$ such that the expected number of tests needed is minimized. Perhaps the simplest example is the classical parlor game "twenty questions", where the objective might be to identify a famous person one player thinks of (the $Y$ in this case) by asking the minimum number questions about $Y$ on average, where each of these questions can be viewed as a test about $Y$.[1] Other examples include Bayesian optimal experimental design (Lindley, 1956), sensor fault detection (Zheng et al., 2012) and medical diagnosis (Peng et al., 2018). Since computing the optimal sequence of tests for such scenarios is NP-complete in general (Hyafil & Rivest, 1976), the "greedy" heuristic of choosing tests in each iteration that reduce the uncertainty about $Y$ the most, given the outcomes observed so far, is commonly employed in practice. More precisely, this is mathematically equivalent to choosing the test whose outcome has maximum mutual information with $Y$ given the sequence of test outcomes observed so far and is popularly known as the Information Maximization (InfoMax) algorithm, which has found numerous uses in recent applications (Geman & Jedynak, 1996; Sznitman & Jedynak, 2010; Branson et al., 2014; Geman et al., 2015; Ma et al., 2018; Foster et al., 2019; Cuturi et al., 2020; He et al., 2022; Chattopadhyay et al., 2022). Given the natural intuition behind InfoMax, one might ask how efficient this greedy heuristic is in practice. However, despite its popularity, theoretical guarantees about the performance of the InfoMax algorithm are scarce (Chen et al., 2015).

In this paper, we analyze the InfoMax algorithm for binary tests and derive bounds on its performance. Throughout this paper, by performance we mean the expected number of tests needed to make accurate predictions. If one has access to all possible binary functions of $Y$ as tests, then it is

---

[1]For example, one possible question could be "Is $Y$ still alive?"

known that the performance of the greedy strategy is upper bounded by $H(Y)+1$ (Garey & Graham, 1974), where $H(Y)$ denotes the entropy of $Y$. This is nearly-optimal since $H(Y)$ is a lower bound on the best possible performance (Shannon, 1948). Unfortunately, for scenarios when one has access to only a restricted set of functions of $Y$, Loveland (1985) illustrated that it is possible to construct binary active testing problems for which given a set of tests, $\mathcal{T}$, the greedy strategy requires at least $\frac{|\mathcal{Y}|}{16} \times \text{opt}(\mathcal{T}, Y)$ number of of tests to identify $Y$, where $|\mathcal{Y}|$ is the number of values $Y$ can take and $\text{opt}(\mathcal{T}, Y)$ is the performance of the optimal (not necessarily greedy) strategy for identifying $Y$ given $\mathcal{T}$. Thus, as $|\mathcal{Y}|$ gets large, the greedy strategy can obtain dismal results when compared with the optimal strategy. In light of this result, how is it that the greedy strategy is one of the most popular heuristics used for solving the sequential selection of tests in practical applications?

In this paper, we argue that the competitive performance of the greedy strategy that is often observed in practice can be attributed to a property of the set of available tests $\mathcal{T}$ that we call $\delta$-unpredictability. A set of tests is $\delta$-unpredictable, if in every iteration of the greedy strategy the selected test has conditional probability of being 'true', given history of test outcomes observed so far, within $\frac{1}{2} \pm$

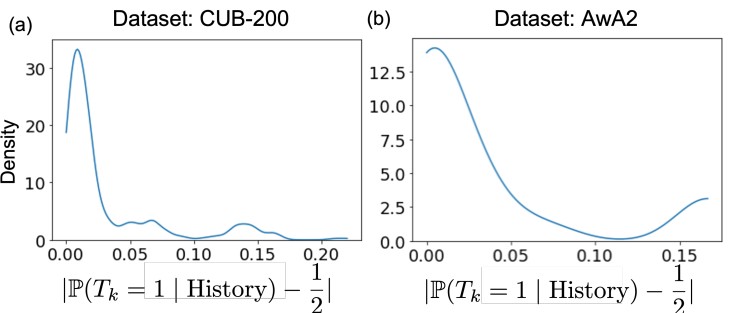

Figure 1: Distribution of the values of $|\mathbb{P}(T_k = 1 \mid \text{History}) - \frac{1}{2}|$ sampled over all iterations for all examples in the dataset, where $T_k$ indicates the test selected at iteration $k$.

$\delta$, unless the posterior over $Y$ given the history of outcomes observed so far is sufficiently peaked, upon which the algorithm will terminate. While taking $\delta = \frac{1}{2}$, makes any given $\mathcal{T}$ trivially $\delta$-unpredictable, we observe that in many practical applications the given set of tests is $\delta$-unpredictable for modest values of $0 \ll \delta \ll \frac{1}{2}$. For example, in Figure 1, we show on two machine learning datasets, namely CUB-200 (Wah et al., 2011) and AwA2 (Xian et al., 2018), on which carrying out information maximization always finds a test within $\delta$ units of one-half in every iteration for every datapoint with $\delta = 0.22$ and $\delta = 0.17$, respectively. More details in Appendix §A.5. Similarly, Geman et al. (2015) employed $\delta$-unpredictable $\mathcal{T}$ for visual scene annotation (in terms of objects in the scene, their attributes and relationships) and showed $\delta = 0.15$ works.

Inspired by these observations, we study the performance of the greedy strategy when $\mathcal{T}$ is $\delta$-unpredictable for some $\delta \in [0, \frac{1}{2}]$. In the extreme case where $\delta = 0$, we have bisecting tests at each iteration. If we further assume the tests are functions of $Y$, then the set of possible values $Y$ can take, referred to as the active set, is effectively halved at each iteration depending on the test outcome. This is akin to binary search, which is known to converge in $H(Y)$ iterations (Flores & Madpis, 1971). On the other extreme, when $\delta = \frac{1}{2}$, it allows the greedy strategy to pick a test that is deterministic given the history, in other words, tests with conditional probability of being true equal to 0/1 will be selected which would lead to null reduction of the current active set. Our contribution, is to study what happens in the middle, say when $\delta \approx 0.25$. We first study the simpler case of oracle tests, that is, when all tests in $\mathcal{T}$ are functions of $Y$ and bound the performance of the greedy strategy to be at most $\frac{H(Y)}{-\log_2(\frac{1}{2}+\delta)}$, which immediately improves upon bounds previously reported in literature (Garey & Graham, 1974; Loveland, 1985; Dasgupta, 2004; Kosaraju et al., 1999). Building on this, we extend our analysis and present our main result on the algorithm's performance under noisy tests. In particular, we assume the test outcomes are corrupted by a binary symmetric channel and we obtain bounds on the prediction error rate after the algorithm terminates. The analysis in the noisy case is more involved since the test outcomes, by virtue of noise, no longer constrain the set of possible values $Y$ can take. In summary, our main contributions are the following.

- We first study the oracle case where tests are functions of $Y$. Assuming the given set of tests, $\mathcal{T}$, is $\delta$-unpredictable for some $\delta \in [0, \frac{1}{2}]$, we prove that the greedy strategy needs at most $\frac{H(Y)}{-\log_2(\frac{1}{2}+\delta)}$ number of tests on average to identify (predict) $Y$. To the best of our knowledge, this is a first bound on the performance of the greedy strategy that explicitly depends on the entropy of $Y$. This

is desirable since a lower bound on the average number of tests needed for any given $\mathcal{T}$ is given by the entropy of $Y$ (Shannon, 1948). Moreover, we show that our bound is tighter than previously known bounds for oracle tests in practically relevant settings.

• We then extend our analysis to the noisy case where we assume that test outcomes are corrupted via a binary symmetric channel. We obtain an upper bound on the performance of the greedy strategy that explicitly depends on $\delta$ and the noise level. Specifically, our bound in this case is again within a constant factor of the entropy of $Y$ modulo an additional term, where the constant factor and the additional term depend on $\delta$ and the noise level. To the best of our knowledge, this is the first such result for the greedy strategy given noisy tests.

## 2 RELATED WORK

Information Maximization (InfoMax) is a popular heuristic for sequentially selecting tests to make accurate predictions, which has been widely adopted across various fields under different names. One of the first proposals of this algorithm was in the context of optimal experimental design by Lindley (1956) where tests correspond to experiments one can carry out to gather information about $Y$. Consequently, this algorithm has been proposed under various names such as Probabilistic Bisection Method, (Horstein, 1963), Splitting Algorithm (Garey & Graham, 1974), Entropy Testing (Geman & Jedynak, 1996), Information Gain (for decision tree induction) (Breiman et al., 1984), Generalized Binary Search (Dasgupta, 2004), and Information Pursuit (Jahangiri et al., 2017). Inspired by its empirical success, there is a fifty year lineage of scattered work on the performance of this "greedy" strategy. We begin by reviewing works studying the oracle case, where tests are functions of $Y$, and conclude by mentioning recent efforts towards analyzing the more general case where test outcomes are corrupted by noise.

**Oracle tests.** Shannon (1948) showed that when $\mathcal{T}$ is complete (that is, we have a test for every function of $Y$), the greedy strategy requires at most one test more than the optimal strategy on average. This result was extended by Sandelius (1961) who showed that greedy is in fact optimal when $Y$ is uniformly distributed. Usually, for practical applications, $\mathcal{T}$ will almost always be incomplete. For example, in the popular "twenty question" parlor game involving famous people, we cannot test if $Y$ is in every possible subset of famous people using questions about presence or absence of single human attributes like "writer", "female", "living", "French", etc. Subsequently Kosaraju et al. (1999) and Dasgupta (2004) proved that in the case of incomplete tests, the greedy strategy would require at most $\mathcal{O}\left(\ln\left(\frac{1}{\min_{y \in \mathcal{Y}} \mathbb{P}(Y=y)}\right) \times \text{opt}(\mathcal{T}, Y)\right)$ number of queries on average. Here $\text{opt}(\mathcal{T}, Y)$ is, as defined in the Introduction, the performance of the optimal strategy for identifying $Y$. This generic bound is often vacuous (too loose) in practice as we also show empirically in the appendix (see §A.5). The idea of assuming the existence of $\delta$-unpredictable tests in each iteration of the greedy strategy was considered in earlier work (Garey & Graham, 1974; Loveland, 1985). However, their analysis technique is significantly different from ours and results in an upper bound of $\frac{\log_2 |\mathcal{Y}|}{|(\frac{1}{2}-\delta) \log_2(\frac{1}{2}-\delta)|} + \frac{1+2\delta}{1-2\delta}$, which is typically larger (i.e., looser) than ours. See §4.2 for an extended discussion comparing the these bounds with our proposed bound.

**Noisy tests.** This refers to the situation where the tests $T$ are not determined by $Y$, that is, the entropy $H(T \mid Y)$ is positive. Unlike the oracle case, the performance of the greedy strategy in this case is sparsely explored. It is known that InfoMax is optimal in the restricted case where $Y \in \mathbb{R}$ and $\mathcal{T}$ is a set of noisy indicator functions for all possible finite unions of intervals along the real line (Jedynak et al., 2012). More general results are obtained by reducing the noisy case to the oracle case. For instance, Nowak (2008) assumed that the tests are "repeatable", that is, any given test can be independently replicated any number of times to obtain the true outcome (de-noise) with high probability. Thus, by repeating the same test multiple times, its outcome can be made deterministic given $Y$ (with high confidence) and the results discussed for the oracle case apply with an additional cost for repeating the test. However, this is not very realistic since in practice we rarely have access to "repeatable" tests. Golovin et al. (2010) analyzed greedy active learning algorithms in the presence of noise by considering the tests to be functions of $Y$ *and* some noise variable $\eta$ with known joint distribution $\mathbb{P}(Y, \eta)$, and thereafter applied the bounds known from the oracle case. Finally, Chen et al. (2015) explored the near-optimality of information maximization for the more practical scenario where noise is persistent, that is, tests are not "repeatable". Compared to our work,

Chen et al. (2015) studies the setting "What is the maximum amount of mutual information one can obtain about $Y$ by carrying out $k$ tests following the greedy strategy?", whereas we are interested in bounding the mean number of tests required to achieve a desired level of accuracy.

## 3 PROBLEM SETTING AND PRELIMINARIES

As is common convention, we will use capital letters for random variables and lowercase letters for their realizations. We will use the symbol $\mathbb{P}(\mathcal{E})$ to denote the probability of event $\mathcal{E}$. Moreover, we will often refer to the Information Maximization (InfoMax) algorithm simply as the *greedy strategy*.

**Information maximization.** InfoMax (Geman & Jedynak, 1996) is a greedy strategy for selecting tests sequentially in order of information gain. More formally, let $Y$ be a discrete random variable taking values in $\mathcal{Y}$ and let $\mathcal{T}$ be a given finite set of available tests, whose outcomes are informative about the value of $Y$. All random variables ($\mathcal{T}$ and $Y$) are defined on a common sample space $\Omega$. Given this setup, for any collection of tests (binary, noisy or otherwise), the InfoMax algorithm proceeds iteratively as follows:

$$T_1 = \arg\max_{T \in \mathcal{T}} I(T; Y); \quad T_{k+1} = \arg\max_{T \in \mathcal{T}} I(T; Y \mid \mathcal{A}(t_{1:k})). \tag{1}$$

Here $T_{k+1} \in \mathcal{T}$ refers to the new test selected by InfoMax at step $k+1$, based on the history of outcomes to previously asked tests (denoted as $t_{1:k}$), and $t_{k+1} \in \{0, 1\}$ indicates the corresponding outcome of the test asked in iteration $k+1$. The conditioning event $\mathcal{A}(t_{1:k})$ is defined as the event $\{\omega \in \Omega : T_i(\omega) = t_i : i \in \{1, 2, \ldots, k\}\}$, where $t_i$ is the observed outcome for carrying out test $T_i$. We refer to these events as *active sets*. We will use the concept of active sets in our analysis of InfoMax. The algorithm terminates after $L$ iterations if either $\max_{y \in \mathcal{Y}} \mathbb{P}(Y = y \mid \mathcal{A}(t_{1:L})) > \gamma$ (a hyper-parameter that can be interpreted as desired accuracy level) or after all tests have been carried out. Refer to Figure 5 in the appendix for a flowchart diagram illustrating the InfoMax algorithm. Having described the InfoMax algorithm, we next define $(\delta, \gamma)$-unpredictable set of tests which encapsulates our assumption of existence of approximately bisecting sets as discussed in the Introduction.

**Unpredictable set of tests.** As motivated in the Introduction, there exists scenarios when the greedy strategy can perform poorly compared to the optimal strategy. This calls for some assumptions on $\mathcal{T}$ to ensure the good performance of the greedy strategy that is often observed in practical sequential testing problems. In this work, we assume that at each iteration of the greedy algorithm there exists a test that $\delta$-approximately bisects the current active set. Formally,

**Definition 1.** [$(\delta, \gamma)$-**unpredictable set of tests**] A set of tests $\mathcal{T}$ is said to be $(\delta, \gamma)$-unpredictable[2] if at any iteration $k+1$ of InfoMax (assuming there remain tests in $\mathcal{T}$ that have not yet been carried out), either

- The probability of the mode of the posterior is greater than or equal to $\gamma$, i.e., $\max_Y \mathbb{P}(Y \mid \mathcal{A}(t_{1:k})) \geq \gamma$; or

- There exists a test $T_{k+1} \in \mathcal{T}$ such that,

$$\left| \mathbb{P}\left(T_{k+1} = 1 \mid \mathcal{A}(t_{1:k})\right) - \frac{1}{2} \right| \leq \delta, \tag{2}$$

where $t_{1:k}$ denotes the history of test outcomes after $k$ iterations.

The $\gamma$ parameter is user-defined and controls the termination criteria for the greedy strategy. In the extreme case where we require $Y$ to be identifiable, $\gamma = 1$. For simplicity, in such scenarios, we will drop $\gamma$ from the notation and refer to such sets as $\delta$-unpredictable set of tests, implicitly meaning that the algorithm terminates only when $Y$ is identified or all tests in $\mathcal{T}$ have been carried out. We will further discuss the motivation for this definition and how it helps us bound the performance of the greedy strategy in the subsequent sections.

---

[2]The word unpredictable comes from the fact that if a test $T' \in \mathcal{T}$ at iteration $k$ exactly bisects the current active set, then, one cannot predict the outcome of $T'$ based on the history of test outcomes observed up till the first $k - 1$ iterations better than a random (unbiased) coin flip.

## 4 PERFORMANCE BOUNDS FOR ORACLE TESTS

In this section, we analyze the performance of InfoMax when all tests in $\mathcal{T}$ are functions of $Y$, hence the name *oracle tests*. Throughout this section, we will denote the outcome of test $T$ as $T(Y)$ to explicitly remind the reader that $T$ is a function of $Y$. Effectively the sample space $\Omega$ (as defined in §3) can be taken to be $\mathcal{Y}$. Since the tests are not noisy, it is reasonable to expect that they collectively determine $Y$, that is, the value of $Y$ is uniquely determined if we observe $\{T(Y), \forall T \in \mathcal{T}\}$). As a result we will drop $\gamma$ from the notation and only refer to $\mathcal{T}$ as being a $\delta$-unpredictable set of tests.

### 4.1 A NEW BOUND ON THE PERFORMANCE OF THE GREEDY INFORMATION MAXIMIZATION ALGORITHM

**Relationship with entropy maximization.** In the oracle case, where $Y$ determines the test outcomes (i.e., the outcome of any test is a function of $Y$, $t = T(Y)$, $\forall T \in \mathcal{T}$), the InfoMax algorithm as described in equation 1 is equivalent to sequentially finding the test $T$ that achieves the maximum conditional entropy given history. Equivalently,

$$T_1 = \arg\max_{T \in \mathcal{T}} H(T); \quad T_{k+1} = \arg\max_{T \in \mathcal{T}} H(T \mid \mathcal{A}(t_{1:k})). \tag{3}$$

The equivalence of equation 3 and equation 1 can be seen by noticing that $H(T \mid Y, \mathcal{A}(t_{1:k})) = 0$ when all tests are functions of $Y$ (Cover, 1999). Note that the active set in this case is now simply a subset of $\mathcal{Y}$, that is, $\mathcal{A}(t_{1:k}) = \{y \in \mathcal{Y} : T_i(y) = t_i : i \in \{1, 2, ..., k\}\}$.

**Motivation for assuming $\mathcal{T}$ is $\delta$-unpredictable.** The motivation for assuming a given $\mathcal{T}$ is $\delta$-unpredictable is as follows. The entropy of a binary random variable is maximized when its success probability is $p = \frac{1}{2}$. Equation 3 can be reinterpreted as sequentially selecting tests from $\mathcal{T}$ that have success probability closest to $\frac{1}{2}$ given the history of test outcomes observed so far. Specifically,

$$T_1 = \arg\min_{T \in \mathcal{T}} \left| \mathbb{P}(T(Y) = 1) - \frac{1}{2} \right|; \quad T_{k+1} = \arg\min_{T \in \mathcal{T}} \left| \mathbb{P}(T(Y) = 1 \mid \mathcal{A}(t_{1:k}) - \frac{1}{2} \right|; \tag{4}$$

While it will generally not be possible to find a perfectly bisecting test, it is reasonable to assume that there exists some $\delta$, such that at any iteration, a test can be found in $\mathcal{T}$ whose success probability, conditioned on the history of test outcomes observed so far, is within $\frac{1}{2} \pm \delta$, as motivated in §1.

**Bounding the performance of the greedy strategy.** If $\mathcal{T}$ is $\delta$-unpredictable for very small $\delta$ we can intuitively expect the number of queries needed on average to identify $Y$ to be roughly of the order of $H(Y)$ (since we have almost bisecting tests). On the other hand, for large $\delta$ (close to $\frac{1}{2}$), any given set of tests would be $\delta$-unpredictable (according to definition 1) and we would expect the number of queries needed on average to blow up. The following theorem captures this intuition and provides a bound on the expected number of tests needed by the greedy strategy as a function of both $\delta$ and the entropy of $Y$.

**Theorem 1.** *Fix any $\delta \in [0, \frac{1}{2}]$. Given a $\delta$-unpredictable $\mathcal{T}$, the average number of tests needed by the information maximization algorithm to identify $Y$ is at most*

$$B_{Ours} := \frac{H(Y)}{-\log_2(\frac{1}{2} + \delta)}. \tag{5}$$

*Proof.* (Sketch only; see Appendix §A.1.1 for a complete proof) Our result is based on the insight that if at any iteration $k$, the greedy strategy picks a test $T_k$ that satisfies equation 2, then at least $\frac{1}{2} - \delta$ of the probability mass of the active set $\mathcal{A}(t_{1:k-1})$ would be discarded depending on the outcome $T_k(Y)$. Applying this argument recursively, the probability mass of the active set after $k$ iterations, $\mathbb{P}(\mathcal{A}(t_{1:k}))$, is at most $(\frac{1}{2} + \delta)^k$. As a result, we can conclude that if $Y = y$ is still in the active set after iteration $k$ then it must be that $\mathbb{P}(Y = y) \leq (\frac{1}{2} + \delta)^k$. This result gives a bound on the number of tests needed to identify state $y$ which is then used to bound the average number of tests. $\square$

To highlight the importance of this result, recall from coding theory that given any set of tests, the optimal strategy cannot be better than $H(Y)$, which thus serves as a lower bound for the greedy strategy given any $\mathcal{T}$. To the best of our knowledge, our result is the first one to upper bound the

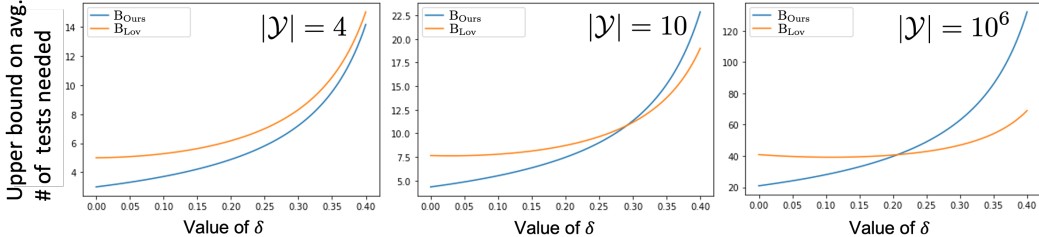

Figure 2: Comparing our bound $B_{\text{Ours}}$ with $B_{\text{Lov}}$ for different values of $|\mathcal{Y}|$ and $\delta$. When $|\mathcal{Y}|$ (the number of discrete values $Y$ can take) is small ($|\mathcal{Y}| = 4$ here), our bound is uniformly a tighter upper bound than the Loveland bound. As $|\mathcal{Y}|$ increases for larger $\delta$ values, $B_{\text{Lov}}$ gets tighter. Asymptotically ($|\mathcal{Y}| \approx 10^6$ here), we see that our bound is tighter for small values of $\delta < 0.2$.

performance of the greedy strategy to be at most a multiplicative factor of the entropy of $Y$. This multiplicative factor degrades inverse logarithmically with $\delta$ and so even for the modest value of $\delta \approx 0.2$, which is far from a bisecting split, our result guarantees that the average number of tests under the greedy strategy is at most roughly twice the entropy of $Y$.

## 4.2 COMPARISON WITH PREVIOUS BOUNDS

Having described our bound, we compare it with bounds previously reported in literature. The assumption of a $\delta$-unpredictable $\mathcal{T}$ was previously considered by Garey & Graham (1974) for the case where $Y$ is uniformly distributed, and subsequently by Loveland (1985) for any distribution on $Y$. Both papers get the same bound and so we compare with the bound in Loveland (1985), which we will refer to as the $B_{\text{Lov}}$. Their analysis technique is significantly different from ours and as a result they obtain a very different upper bound on the average number of queries needed to identify $Y$,

$$B_{\text{Lov}} := \frac{\log_2 |\mathcal{Y}|}{-(\frac{1}{2} - \delta) \log_2(\frac{1}{2} - \delta)} + \frac{1 + 2\delta}{1 - 2\delta}, \tag{6}$$

where $|\mathcal{Y}|$ is the number of discrete values $Y$ can take. Comparing the bound in equation 6 with our bound in equation 5 we make the following observations.

- When $Y$ is uniform, we can compare the two bounds more easily since $H(Y) = \log_2 |\mathcal{Y}|$. As illustrated in figure 2, when $|\mathcal{Y}|$ is small our bound is uniformly tighter than $B_{\text{Lov}}$ for all values of $\delta \in [0, \frac{1}{2}]$. As $|\mathcal{Y}|$ increases, $B_{\text{Lov}}$ gets tighter for larger values of $\delta$. Asymptotically, when $\delta \to \infty$, our bound is tighter whenever $\delta \leq 0.1963 \approx 0.2$. This is a favorable result since we are most interested in the regime where $\delta$ is moderate (0.15-0.2), because otherwise the greedy strategy can degrade significantly compared to the optimal strategy. Indeed, note that both bounds diverge to $\infty$ for larger values of $\delta$.

- When $Y$ is uniform and $\delta \to 0$, that is when we have access to exactly bisecting splits of the current active set, our bound recovers the entropy bound (Shannon, 1948) of $\log_2 |\mathcal{Y}|$, whereas $B_{\text{Lov}}$ converges to $2 \log_2 |\mathcal{Y}| + 1$, indicating a clear gap.

- When the distribution for $Y$ is not uniform, we expect our bound to be tighter since it depends on $H(Y)$ instead of $\log_2 |\mathcal{Y}|$. More specifically, for large $|\mathcal{Y}|$, the value of $\delta$ at which the two bounds are equal increases from $\approx 0.2$ (which was the point at which both the bounds were equal in the uniform case, refer Figure 2 for the case $|\mathcal{Y}| = 10^6$) as the level of non-uniformity of $Y$ increases. This is illustrated in Figure 3.

Next we compare $B_{\text{Ours}}$ with the bounds derived by Dasgupta (2004) and Kosaraju et al. (1999), which make no assumption on $\mathcal{T}$. Since both papers have the same bound in big $\mathcal{O}$ notation, we compare with Dasgupta's bound because it specifies the constants explicitly. Let $\text{opt}(\mathcal{T}, Y)$ be the expected number of queries needed by the optimal strategy to identify $Y$ given a set of tests $\mathcal{T}$. Dasgupta's bound is given by:

$$B_{\text{Das}} := 4 \ln \left( \frac{1}{\min_Y p(Y)} \right) \times \text{opt}(\mathcal{T}, Y). \tag{7}$$

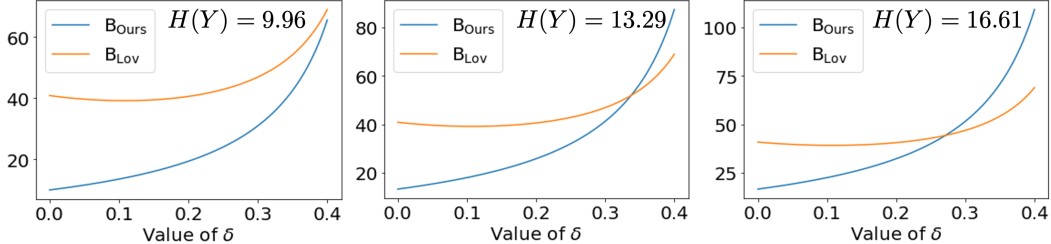

Figure 3: Comparing our bound $B_{\text{Ours}}$ with $B_{\text{Lov}}$ for different values of $H(Y)$ and $\delta$, with $|\mathcal{Y}| = 10^6$. If $Y$ was uniform $H(Y)$ in this case would be $\log_2(10^6) = 19.93$ bits. We see that as the entropy decreases, the value of $\delta$ at which the two bounds are equal increases from $\approx 0.2$ for the uniform distribution (Figure 2 col 3) to about $0.4$ for when the distribution over $Y$ has $H(Y) = 9.96$ bits.

Notice that unlike the previous bounds, $B_{\text{Das}}$ depends on $\text{opt}(\mathcal{T}, Y)$. This is because, in absence of any assumption on $\mathcal{T}$ (like the $\delta$-uncertainty assumption we make), it only makes sense to analyze the performance of the greedy strategy relative to that of the optimal strategy. Otherwise, one can always choose some inefficient $\mathcal{T}$ to make the greedy strategy perform arbitrarily bad. For example, take $\mathcal{T}$ to contain only singleton tests of the form "Is Y = y?".

Using the fact that $\text{opt}(\mathcal{T}, Y) \geq H(Y)$ Shannon (1948) and $\ln\left(\frac{1}{\min_Y p(Y)}\right) \geq \ln|\mathcal{Y}|$, we can show that for $\delta \leq 2^{-\frac{1}{4\ln|\mathcal{Y}|}} - \frac{1}{2}$, our bound is guaranteed to be tighter that $B_{\text{Das}}$. For details see Appendix §A.4. This is expected since $B_{\text{Das}}$ makes no distributional assumptions about $\mathcal{T}$. We explicitly evaluate $2^{-\frac{1}{4\ln|\mathcal{Y}|}} - \frac{1}{2}$ for values of $|\mathcal{Y}| \in [10, 100]$ and show in Figure 4 that our bound is tighter for extremely modest values of $\delta \leq 0.43$ (recall $\delta \in [0, 0.5]$).

We demonstrate on two machine learning datasets (CUB-200 (Wah et al., 2011) and AwA2 (Xian et al., 2018)) that the given set of tests $\mathcal{T}$ is $\delta$-unpredictable for modest values of $\delta$ (0.22 and 0.17 respectively) and subsequently show that our bound is closer to the true mean number of tests the greedy strategy requires on these datasets to identify $Y$, than the other discussed bounds. These results can be found in Appendix §A.5.

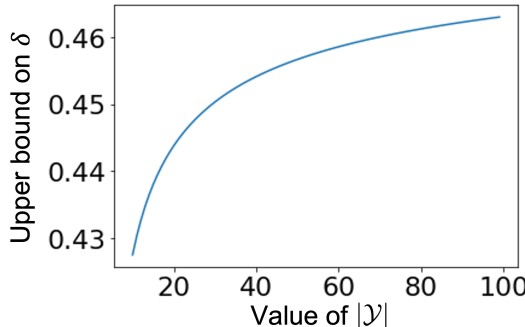

Figure 4: Comparing our bound with $B_{\text{Das}}$ by evaluating $2^{-\frac{1}{4\ln|\mathcal{Y}|}} - \frac{1}{2}$ for different values of $|\mathcal{Y}|$. The y-axis is an upper bound on the value of $\delta$ for which our bound will be tighter than Dasgupta's.

## 5 PERFORMANCE BOUNDS FOR NOISY TESTS

Here, we analyze the performance of the greedy strategy when all tests in $\mathcal{T}$ are noisy, that is $\forall T \in \mathcal{T}$, the conditional entropy $H(T \mid Y) > 0$. As discussed in §2, the performance of the greedy strategy under noise is poorly understood. Unlike prior work Nowak (2008), our analysis does not assume that tests can be repeated any number of times to average the noise out. This is because in many applications the same test cannot be repeated again or will give the same outcome Chen et al. (2015).[3] Instead we consider an explicit noise model for the tests and analyze the performance of the greedy strategy for that model.

---

[3]Note, while we do not assume the same test can be repeated, there can be multiple tests in $\mathcal{T}$ that are (conditionally) statistically identical. For example, in the famous 20Q game let $y_1$ = "Queen Victoria" and $y_2$ = "Charles Darwin" be the only two states with non-trivial mass. Then, both tests "Is Y female?" and "Is Y a queen?" have statistically identical outcomes but are different tests.

**Binary Symmetric Channel (BSC) Noise Model.** We first study the case where test outcomes are corrupted by a BSC, which is perhaps the most well-studied and simplest model for understanding the effects of noise in communication channels (Shannon, 1948). We make the following assumptions.

- For every $T \in \mathcal{T}$ there exists random variables $D_T(Y)$, which is a function of $Y$, and $N_T$ such that $T = D_T(Y) \oplus N_T$. The symbol $\oplus$ denotes the Exclusive OR (XOR) operation. $D_T(Y)$ can be understood as the true outcome for test $T$ if there was no noise. $N_T$ is the noise variable that corrupts the test outcome.

- For every $T \in \mathcal{T}$, we assume $N_T$ is independent of $Y$ with prior probability $\mathbb{P}(N_T = 1) = \alpha$ for some $\alpha \in [0, \frac{1}{2}]$. Moreover, we assume all the noise variables, $\{N_T : T \in \mathcal{T}\}$, are independent and hence the noise variables are i.i.d..

We now describe our analysis of how the greedy strategy performs under this noise model.

## 5.1 A BOUND OF THE PERFORMANCE OF THE GREEDY STRATEGY FOR NOISY TESTS

In general, when noise is present in the test outcomes, InfoMax (equation 1) is not equivalent to entropy maximization (equation 3). As a result we cannot interpret the greedy strategy as selecting the test at each iteration whose success probability given the history of test outcomes observed so far is close to $\frac{1}{2}$. However, as we show in Lemma 2, under our noise model, we can interpret the greedy strategy as choosing the test $\hat{T}$ in each iteration whose true outcome ($D_{\hat{T}}$) has success probability (given history) closest to a half. We now state our lemma which is inspired from Jedynak et al. (2012) where a similar result was derived for the case where $Y = \mathbb{R}$ and the tests are unions of intervals along $\mathbb{R}$.

**Lemma 2.** *Under the BSC noise model, at any iteration $k + 1$, the InfoMax algorithm will pick test*

$$T_{k+1} = \arg \min_{T \in \mathcal{T}} \left| \mathbb{P}(D_T = 1 \mid \mathcal{A}(t_{1:k})) - \frac{1}{2} \right|,$$

*where $\mathcal{A}(t_{1:k})$ is the active set after $k$ iterations.*

The lemma is proved using standard information-theoretic identities coupled with the properties of our noise model. Refer Append §A.1.2 for a detailed proof. This result is in line with intuition since the noisy component of every test ($N_T$) is independent of $Y$ and hence uninformative for prediction. Thus, the selection of the most information next test is governed solely by how well it's true outcome approximately bisects the current active set $\mathcal{A}(t_{1:k})$.

A natural question to ask next is, *If a given set of tests $\mathcal{T}$ is $(\delta, \gamma)$-unpredictable then what can we conclude about the chosen test's $\mathbb{P}(D_{T_{k+1}} = 1 \mid \mathcal{A}(t_{1:k}))$?* The following lemma answers this.

**Lemma 3.** *Under the BSC model with noise parameter $\alpha \in [0, \frac{1}{2}]$, if $\mathcal{T}$ is $(\delta, \gamma)$-unpredictable according to definition 1, then in any iteration $k + 1$, the greedy strategy will either choose a test $T_{k+1} \in \mathcal{T}$ such that*

$$\left| \mathbb{P}\left( D_{T_{k+1}} = 1 \mid \mathcal{A}(t_{1:k}) \right) - \frac{1}{2} \right| \leq \frac{\delta}{1 - 2\alpha}, \tag{8}$$

*or terminate according to $\gamma$ stopping criterion. Moreover, given $\alpha$, it is not possible to have a $(\delta, \gamma)$-unpredictable $\mathcal{T}$ for $\delta > \frac{1}{2} - \alpha$.*

Refer to Appendix §A.2 for a proof. The above result has two consequences.

1. It shows that for a fixed $\delta$, as the noise level $\alpha$ increases from 0 to $\frac{1}{2}$ (it's maximum possible value) the ability of the true outcome $D_T = D_T(Y)$ for any given test $T \in \mathcal{T}$ to approximately bisect the current active set detoriates by a factor of $\frac{1}{1-2\alpha}$ compared to the observed test outcome $T = t$. Based on this, one can conjecture that as the noise level increases, more and more tests would be needed to identify $Y$, because the ability of the true outcomes to approximately bisect the current active set degrades.

2. It shows that the maximum possible value of $\delta$ is bounded by the noise level $\alpha$. In particular, by inverting the result in equation 8 (see Appendix §A.2) we see that if $\left| \mathbb{P}\left( D_{T_{k+1}} = 1 \mid \mathcal{A}(t_{1:k}) \right) - \right.$

$\frac{1}{2}\Big| \leq \delta'$ for some constant $\delta' \in [0, \frac{1}{2}]$, then this implies $\delta = \delta'(1 - 2\alpha) \in [0, \frac{1}{2} - \alpha]$. Thus, unlike the oracle case, it is not possible to have a set of noisy tests which is $(\delta, \gamma)$-unpredictable for $\delta > \frac{1}{2} - \alpha$. In hindsight, this result makes sense since according to our noise model, every test outcome is corrupted independently of all other tests and hence, there will always be some uncertainty in a certain test's outcome regardless of how many tests have been carried out so far.

Having stated all the ingredients we will now present our main result for the greedy strategy under the BSC noise model.

**Theorem 4.** *Fix noise level $\alpha \in [0, \frac{1}{2}]$ for the BSC model. Fix $\delta \in [0, \frac{1}{2} - \alpha]$. Given a $(\delta, \gamma)$-unpredictable $\mathcal{T}$, the average number of tests needed by the InfoMax algorithm to predict $Y$ with confidence at least $\gamma$ under the BSC model is at most*

$$B_{Ours}^{Noisy} := \frac{H(Y) - |\log_2 \gamma| + \alpha|\mathcal{T}|\log_2 \frac{1-\alpha}{\alpha}}{\log_2(1-\alpha) - \log_2(\frac{1}{2} + \delta)} + 1. \tag{9}$$

A complete proof can be found in Appendix §A.3. The main idea behind the proof of this theorem is similar to the proof sketch for Theorem 1 but requires a more careful tracking of how much probability mass of the active set is discarded in each iteration. As expected, the number of tests needed increases as the desired accuracy level $\gamma$ is increased. Observe that in the absence of noise, that is, when $\alpha = 0$, and we set our desired accuracy to $\gamma = 1$, we recover our bound for the oracle case (Theorem 1)[4]

Our bound guarantees that as long as the noise level $\alpha$ is low relative to the entropy of $Y$, the performance of the greedy strategy is nearly-optimal (that is, within a constant factor of $Y$). To the best of our knowledge, this is a first such result for the InfoMax algorithm for noisy tests. Notice that this bound also depends on $|\mathcal{T}|$ which has a detrimental effect on the bound when the noise level $\alpha$ is high. This dependence on the size of the set of available tests is expected under the BSC model since there will always exist some sample points along which $Y$ cannot be predicted with $\gamma$ level of confidence. As a result, for those sample points, the greedy strategy would end up exhausting all $|\mathcal{T}|$ tests. In the extreme case, when $\alpha = \frac{1}{2}$, none of the tests are informative about $Y$, and hence $Y$ can never be identified in less than $|\mathcal{T}|$ number of tests.

## 6    Conclusion & Limitations

We analyzed the Information Maximization (InfoMax) algorithm and derived new upper bounds for the average number of tests needed to predict $Y$. Our results are based on the observation that in most practical applications of InfoMax, one often has access to tests whose outcome partitions the current active set into two sets of sizes between $\frac{1}{2} \pm \delta$ for $0 \ll \delta \ll \frac{1}{2}$. Using this assumption we obtained better bounds for the greedy strategy than previously established in the literature for the case of oracle tests, that is when the tests are functions of $Y$. Subsequently, we extended our results to the case of noisy tests by assuming that the test outcomes are corrupted by a Binary Symmetric Channel and obtain bounds on the performance of the InfoMax algorithm.

We now describe a few limitations of this work. Our analysis assumes tests are $\delta$-unpredictable for modest values of $\delta$, however *a priori* it is not known how to find $\delta$ such that the given set of tests would be $\delta$-unpredictable. Moreover, the BSC noise model assumes i.i.d. noise, however in practice noise is often dependent on the value of $Y$, and test outcomes are often not independent of each other. We would address these limitations in future work by studying more complex noise models and designing testable conditions to verify if a given $\mathcal{T}$ is $\delta$-unpredictable for a given value of $\delta$ or not.

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

# A APPENDIX

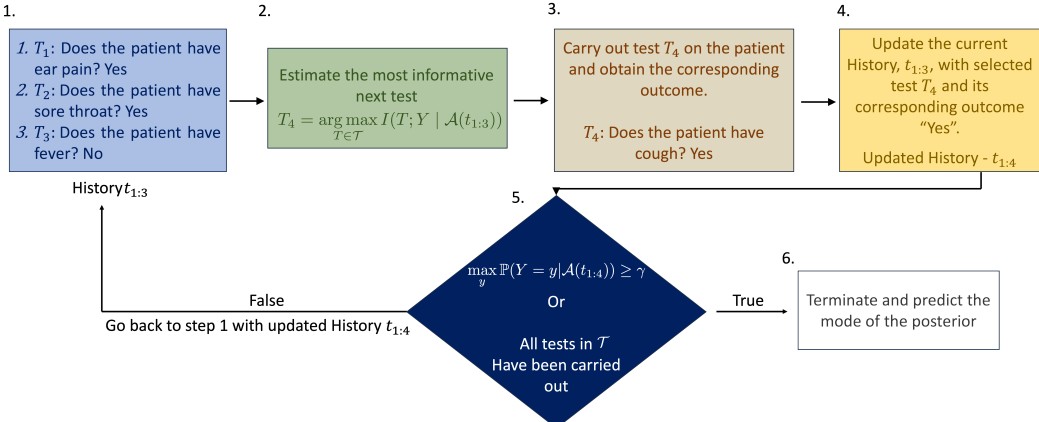

Figure 5: **Illustration of the greedy Information Maximization algorithm.** A flow chart depicting one iteration (k = 4) of the algorithm. As an example we consider the task of disease diagnosis where $Y$ denotes the disease a patient is suffering from (for example, tuberculosis, common cold etc.) and the set of tests $\mathcal{T}$ here corresponds to questions about different symptoms the patient may be experiencing. The corresponding binary answer (Yes/No) to every test indicates the outcome of that test.

## A.1 PROOFS

For ease of reading, we rewrite all the theorems, lemmas and corollaries from the main paper in this section (unnnumbered) before presenting their respective proofs.

### A.1.1 PROOF OF THEOREM 1

**Theorem.** *Fix any $\delta \in [0, \frac{1}{2}]$. Given a $\delta$-unpredictable $\mathcal{T}$, the average number of tests needed by the information maximization algorithm to identify $Y$ is at most*

$$B_{Ours} := \frac{H(Y)}{-\log_2(\frac{1}{2} + \delta)}. \tag{10}$$

The proof of the theorem relies on the following Lemma. Recall from §4.1 that in the oracle case, the active set is simply a subset of $\mathcal{Y}$, that is, $A(t_{1:k}) = \{y \in Y : T_i(Y) = T_i(y) : i \in 1, 2, ..., k\}$.

**Lemma 5.** *If $\mathcal{T}$ is $\delta$-unpredictable for some $\delta \in [0, \frac{1}{2}]$, then after $k$ iterations of the greedy strategy, the probability mass of the current active set $\mathcal{A}(t_{1:k})$ is upper bounded by $(\frac{1}{2} + \delta)^k$.*

*Proof.* We prove this by induction.

**Base case k = 1**. Let $T_1 \in \mathcal{T}$ be the test selected by the greedy strategy in the first iteration. Since $\mathcal{T}$ is $\delta$-unpredictable, we know the new active set $A_{t_1}$, based on the outcome for $T_1$, has mass at most $\frac{1}{2} + \delta$. Thus, $\mathbb{P}(A_{t_1}) \leq \frac{1}{2} + \delta$.

**Base case k = n**. Assume $\mathbb{P}(A_{t_{1:n-1}})$, the probability mass of the active set reached after the first $n-1$ iterations of the greedy strategy, is at most $(\frac{1}{2} + \delta)^{n-1}$. Let $T_n \in \mathcal{T}$ be the test selected by the greedy strategy in the $n^{\text{th}}$ iteration. Since $\mathcal{T}$ is $\delta$-unpredictable, we know the new active set $A_{t_{1:n}}$, based on the outcome for $T_n = t_n$, has probability mass

$$\mathbb{P}(A_{t_{1:n}}) = \mathbb{P}(T_n = t_n \mid A_{t_{1:n-1}})\mathbb{P}(A_{t_{1:n-1}}) \leq (\frac{1}{2} + \delta)(\frac{1}{2} + \delta)^{n-1} = (\frac{1}{2} + \delta)^n,$$

where the first factor in the inequality comes from the assumption that $\mathcal{T}$ is $\delta$-unpredictable, while the second factor is a consequence of the induction hypothesis. $\square$

We will now use the above lemma in the proof of Theorem 1.

*Proof.* For a given $Y = y$, let $l_y$ be the number of iterations the greedy strategy takes before termination, that is after $l_y$ iterations, $y$ has been identified. This implies the current active set $\mathcal{A}(t_{1:l_y}) = \{Y = y\}$. Using Lemma 5 we conclude that

$$
\mathbb{P}(Y = y) \leq (\frac{1}{2} + \delta)^{l_y}
$$
$$
\implies l_y \leq \frac{-\log_2 \mathbb{P}(Y = y)}{|\log_2(\frac{1}{2} + \delta)|}
$$
(11)

Let $l_Y$ now be a random variable denoting the number of tests the greedy strategy requires for random variable $Y$. The expected number of tests needed to identify $Y$ can then be upper bounded as follows,

$$
\mathbb{E}_Y[l_Y] \leq \mathbb{E}_Y\left[\frac{-\log_2 \mathbb{P}(Y = y)}{|\log_2(\frac{1}{2} + \delta)|}\right] = \frac{H(Y)}{-\log_2(\frac{1}{2} + \delta)}.
$$
(12)

In the inequality we used the result from equation 11. $\qquad\square$

As stated in the main paper (see §4.2) our bound, as expressed in equation 5, is tighter than existing bounds on the average number of tests the greedy strategy needs to identify $Y$. Notably, our bound is the first to show that the performance comes within a constant factor of $H(Y)$, the entropy of $Y$. It turns out that, via information-theoretic arguments, this constant factor could be made smaller. This result (which is unpublished) was provided to us during the reviewing process of an earlier version of this paper by an anonymous reviewer. We now state the reviewer's result for completeness.

**Theorem 6.** *Fix any $\delta \in [0, \frac{1}{2}]$. Let $h$ be the binary entropy function. Given a $\delta$-unpredictable $\mathcal{T}$, the average number of tests needed by the information maximization algorithm to identify $Y$ is at most $\frac{H(Y)}{h(\frac{1}{2} + \delta)}$.*

*Proof.* Assume the InfoMax algorithm has run for $k$ iterations. There are two cases,

**Case 1.** $Y$ has not been identified. Then, since $\mathcal{T}$ is $\delta$-unpredictable we know for the test $T_{k+1}$ selected in iteration $k + 1$

$$
H(T_{k+1} \mid T_1 = t_1, \ldots, T_k = t_k) \geq h(\frac{1}{2} + \delta)
$$
(13)

The inequality is obtained using the concavity of the binary entropy function since the chosen test satisfies $|\mathbb{P}(T_{k+1} = 1 \mid T_1 = t_1, \ldots, T_k = t_k) - \frac{1}{2}| \leq \delta$ (from definition 2).

**Case 2.** $Y$ has been identified, at which point the algorithm would terminate and all further tests would have 0 conditional entropy since their outcome will be determined by the value of $Y$ that has been identified, that is,

$$
H(T_{k+1} \mid T_1 = t_1, \ldots, T_k = t_k) = 0
$$
(14)

Define $\tau$ to be the random variable indicating the stopping time for a single run of the InfoMax algorithm. Equation 13 and equation 14 can be combined into

$$
H(T_{k+1} \mid T_1 = t_1, \ldots, T_k = t_k) \geq h(\frac{1}{2} + \delta)\mathbb{1}(\tau > k),
$$
(15)

where $\mathbb{1}$ is the indicator random variable.

Taking expectation on both sides,

$$
H(T_{k+1} \mid T_1, T_2, ..., T_k) \geq h(\frac{1}{2} + \delta)\mathbb{P}(\tau > k),
$$
(16)

Summing $k$ from 0 to $|\mathcal{T}|$ we obtain

$$
\begin{aligned}
h(\frac{1}{2} + \delta) \sum_{k=0}^{|\mathcal{T}|} \mathbb{P}(\tau > k) &\le H(T_1, T_2, ...) \\
&\le H(Y) \\
\implies \mathbb{E}[\tau] &\le \frac{H(Y)}{h(\frac{1}{2} + \delta)},
\end{aligned}
\tag{17}
$$

which is the desired bound. The second inequality is obtained since all the test outcomes are functions of $Y$. $\qquad\square$

### A.1.2 Proof of Lemma 2

**Lemma.** *Under the BSC noise model, at any iteration $k+1$, the information maximization algorithm will pick the test $T_{k+1}$, such that*

$$
T_{k+1} = \arg\min_{T \in \mathcal{T}} \left| \mathbb{P}(D_T = 1 \mid \mathcal{A}(t_{1:k})) - \frac{1}{2} \right|,
$$

*where $\mathcal{A}(t_{1:k})$ is the active set after $k$ iterations.*

*Proof.* Let $h$ be the binary entropy function. At any iteration $k + 1$, the mutual information for any $T \in \mathcal{T}$ can be written as follows,

$$
\begin{aligned}
&I(T, Y \mid \mathcal{A}(t_{1:k}))) \\
&= H(T \mid \mathcal{A}(t_{1:k}))) - H(T \mid Y, \mathcal{A}(t_{1:k}))) \\
&= h\left( \sum_{y \in \mathcal{Y}} \mathbb{P}(y \mid \mathcal{A}(t_{1:k})) \mathbb{P}(T = 1 \mid y) \right) - \sum_{y \in \mathcal{Y}} \mathbb{P}(y \mid \mathcal{A}(t_{1:k})) h\left( \mathbb{P}(T = 1 \mid y) \right)
\end{aligned}
\tag{18}
$$

Define $\delta_T := \sum_{\{y \in \mathcal{Y}: D_T(y)=1\}} \mathbb{P}(y \mid \mathcal{A}(t_{1:k}))$, that is, $\delta_T$ is the total posterior mass on $Y$ (given the current active set) subject to the constraint that $D_T(Y) = 1$. We can rewrite equation 18 as,

$$
\begin{aligned}
I(T, Y \mid \mathcal{A}(t_{1:k}))) &= h\left( \delta_T(1 - \alpha) + (1 - \delta_T)\alpha \right) - \delta_T h(1 - \alpha) - (1 - \delta_T)h(\alpha) \\
&= h\left( \delta_T(1 - \alpha) + (1 - \delta_T)\alpha \right) - h(\alpha).
\end{aligned}
\tag{19}
$$

Equation 19 shows that the mutual information between $T$ and $Y$ given the current active set is just the mutual information of a noisy binary symmetric channel, for which we know the maximal value is attained at $\delta_T = \frac{1}{2}$. Moreover, the binary entropy function is concave thus proving the desired lemma.

$\qquad\square$

### A.2 Proof of Lemma 3

**Lemma.** *Under the BSC model with noise parameter $\alpha \in [0, \frac{1}{2}]$, if $\mathcal{T}$ is $(\delta, \gamma)$-unpredictable according to definition 1, then in any iteration $k + 1$, the greedy strategy will either choose a test $T_{k+1} \in \mathcal{T}$ such that*

$$
\left| \mathbb{P}\left( D_{T_{k+1}} = 1 \mid \mathcal{A}(t_{1:k}) \right) - \frac{1}{2} \right| \le \frac{\delta}{1 - 2\alpha},
\tag{20}
$$

*or terminate according to $\gamma$ stopping criterion. Moreover, given $\alpha$, it is not possible to have a $(\delta, \gamma)$-unpredictable $\mathcal{T}$ for $\delta > \frac{1}{2} - \alpha$.*

*Proof.* We know from the BSC noise model that

$$\mathbb{P}(T_{k+1} = 1 \mid \mathcal{A}(t_{1:k})) = \mathbb{P}(T_{k+1} = 1 \mid D_{T_{k+1}} = 1)\mathbb{P}(D_{T_{k+1}} = 1 \mid \mathcal{A}(t_{1:k}))$$
$$+ \mathbb{P}(T_{k+1} = 1 \mid D_{T_{k+1}} = 0)\mathbb{P}(D_{T_{k+1}} = 0 \mid \mathcal{A}(t_{1:k})). \tag{21}$$

Let $x := \mathbb{P}(D_{T_{k+1}} = 1 \mid \mathcal{A}(t_{1:k}))$. The above equation can be rewritten as

$$\mathbb{P}(T_{k+1} = 1 \mid \mathcal{A}(t_{1:k})) = \mathbb{P}(T_{k+1} = 1 \mid D_{T_{k+1}} = 1)x + \mathbb{P}(T_{k+1} = 1 \mid D_{T_{k+1}} = 0)(1 - x). \tag{22}$$

From our noise model we know $\mathbb{P}(T_{k+1} = 1 \mid D_{T_{k+1}} = 1) = 1 - \alpha$ and $\mathbb{P}(T_{k+1} = 1 \mid D_{T_{k+1}} = 0) = \alpha$. Substituting this in equation 22 we get,

$$\mathbb{P}(T_{k+1} = 1 \mid \mathcal{A}(t_{1:k})) = (1 - \alpha)x + (1 - x)\alpha$$
$$\implies x = \frac{\mathbb{P}(T_{k+1} = 1 \mid \mathcal{A}(t_{1:k})) - \alpha}{1 - 2\alpha} \tag{23}$$

Since $\mathcal{T}$ is $(\delta, \gamma)$-unpredictable, the chosen test $T_{k+1}$ at iteration $k + 1$ satisfies

$$\frac{1}{2} - \delta \leq \mathbb{P}(T_{k+1} = 1 \mid \mathcal{A}(t_{1:k})) \leq \frac{1}{2} + \delta \tag{24}$$

Combining equation 23 and equation 24 we obtain,

$$\frac{\frac{1}{2} - \alpha}{1 - 2\alpha} - \frac{\delta}{1 - 2\alpha} \leq \quad x \quad \leq \frac{\frac{1}{2} - \alpha}{1 - 2\alpha} + \frac{\delta}{1 - 2\alpha}$$
$$\implies -\frac{\delta}{1 - 2\alpha} \leq x - \frac{1}{2} \leq \frac{\delta}{1 - 2\alpha}, \tag{25}$$

proving the lemma.

Moreover, notice that equation 25 can be inverted. In particular if we assume $x := \mathbb{P}(D_{T_{k+1}} = 1 \mid \mathcal{A}(t_{1:k}))$ is between $\frac{1}{2} \pm \delta'$ for some $\delta' \in [0, \frac{1}{2}]$, we can conclude from equation 23 that

$$\left| \mathbb{P}(T = 1 \mid \mathcal{A}(t_{1:k})) - \frac{1}{2} \right| \leq \delta'(1 - 2\alpha). \tag{26}$$

It is immediate from equation 26 that $\mathbb{P}(T = 1 \mid \mathcal{A}(t_{1:k}))$ cannot be more than $\frac{1}{2} - \alpha$ units away from one-half (obtained by setting $\delta' = \frac{1}{2}$ in the above inequality).

The result in equation 26 will be used to reparameterize $\delta$ for a given $(\delta, \gamma)$-unpredictable set of tests in terms of $\delta'$ to get rid of the constraint on $\delta$ in terms of the noise level $\alpha$. $\qquad \square$

### A.3 PROOF OF THEOREM 4

**Theorem.** *Fix noise level $\alpha \in [0, \frac{1}{2}]$ for the BSC model. Fix $\delta \in [0, \frac{1}{2} - \alpha]$. Given a $(\delta, \gamma)$-unpredictable $\mathcal{T}$, the average number of tests needed by the InfoMax algorithm to predict $Y$ with confidence at least $\gamma$ under the BSC model is at most*

$$B_{Ours}^{Noisy} := \frac{H(Y) - |\log_2 \gamma| + \alpha|\mathcal{T}| \log_2 \frac{1-\alpha}{\alpha}}{\log_2(1 - \alpha) - \log_2(\frac{1}{2} + \delta)} + 1. \tag{27}$$

Throughout the remainder of this exposition, we will assume that $\mathcal{T}$ is $(\delta, \gamma)$-unpredictable for some $\delta \in [0, \frac{1}{2} - \alpha]$ and $\gamma \in [0, 1]$. Proof of this theorem will rely on the following lemma.

**Lemma 7.** *Under the BSC model with noise parameter $\alpha$, for any hypothesis $y^0 \in \mathcal{Y}$ with prior probability $p_{y^0}$, if Info-Max has run for $k$ iterations, with $m < k$ iterations for which the observed test outcome $t_i$ is not equal to the corresponding true outcome $D_{T_i}(y^0)$, then*

$$P(y^0 \mid \mathcal{A}(t_{1:k})) \geq \frac{p_{y^0}\alpha^m(1 - \alpha)^{k-m}}{(\frac{1}{2} + \delta)^k}.$$

*Proof.* If the hypothesis of the lemma is true then the joint probability distribution of the observed test outcomes and $y^0$ can be written as

$$\mathbb{P}(Y = y^0, T_1 = t_1, ..., T_k = t_k) = p_{y^0} \alpha^m (1-\alpha)^{k-m}. \tag{28}$$

This is because, given $Y = y^0$, all the test outcomes are conditionally independent of each other and so their probability only depends on whether or not the true outcome was corrupted by noise. Since we know the probability of corruption is $\alpha$ (uniformly for all the tests) we obtain the desired result in equation 28. Using this we can obtain a lower bound on the conditional probability of $y^0$ given the current active set after $k$ iterations:

$$\mathbb{P}(y^0 \mid \mathcal{A}(t_{1:k})) = \frac{p_{y^0} \alpha^m (1-\alpha)^{k-m}}{\mathbb{P}(\mathcal{A}(t_{1:k}))} \geq \frac{p_{y^0} (1-\alpha)^{k-m} (\alpha)^m}{(\frac{1}{2} + \delta)^k}.$$

The inequality is obtained by appealing to Lemma 5. $\qquad \square$

We will now prove our main result (Theorem 4).

*Proof.* For any sample point $\omega \in \Omega$. Let $y^0 = Y(\omega)$ with $\mathbb{P}(Y = y^0) = p_{y^0}$. If the greedy strategy has run for $k$ iterations and not terminated, and there are $m < k$ iterations where the observed test outcome $t_i$ is not equal to the corresponding true outcome $D_{T_i}(y^0)$, then we can bound $k$ as,

$$\gamma \geq \mathbb{P}(y^0 \mid \mathcal{A}(t_{1:k})) \geq \frac{p_{y^0} (1-\alpha)^{k-m} (\alpha)^m}{(\frac{1}{2} + \delta)^k}. \tag{29}$$

This first inequality is because if $P(y^0 \mid \mathcal{A}(t_{1:k})) > \gamma$, then the algorithm will terminate since this would imply $\max_Y P(Y \mid \mathcal{A}(t_{1:k})) > \gamma$. The second inequality is obtained from Lemma 7.

Rearranging equation 29 we get

$$\gamma (\frac{1}{2} + \delta)^k \geq p_{y^0} (1-\alpha)^{k-m} (\alpha)^m. \tag{30}$$

Taking log on both sides,

$$\log_2 \gamma + k \log_2 (\frac{1}{2} + \delta) \geq \log_2(p_{y^0}) + k \log_2(1-\alpha) + m \log_2 \frac{\alpha}{1-\alpha}$$

$$\implies k \left[ |\log_2(\frac{1}{2} + \delta)| - |\log_2(1-\alpha)| \right] \leq -\log_2 p_{y_0} - |\log_2 \gamma| + m \log_2 \frac{1-\alpha}{\alpha} \tag{31}$$

$$\implies k \leq \frac{-\log_2 p_{y_0} - |\log_2 \gamma| + m \log_2 \frac{1-\alpha}{\alpha}}{\left[ |\log_2(\frac{1}{2} + \delta)| - |\log_2(1-\alpha)| \right]}.$$

The last inequality effectively upper bounds $k$ for fixed $m$ number of false answers, which now will be made precise.

For any sample point $\omega \in \Omega$ let $l(\omega)$ be the number of tests the greedy strategy carries out before termination, that is, before the mode of the posterior has mass greater than or equal to $\gamma$ or all $|\mathcal{T}|$ tests have been carried out. Let $m(\omega)$ be the number of iterations for which the observed test outcome does not equal the corresponding true outcome. Using equation 31 we can upper bound $l(\omega)$ as

$$l(\omega) \leq \frac{-\log_2 p_{y_0} - |\log_2 \gamma| + m(\omega) \log_2 \frac{1-\alpha}{\alpha}}{\left[ |\log_2(\frac{1}{2} + \delta)| - |\log_2(1-\alpha)| \right]} + 1. \tag{32}$$

The +1 term shows up because one more test is needed to guarantee the mode of the posterior given history would have probability more than $\gamma$. However, in the case when $\gamma = 1$, we note that since the probability can never be more than 1, this +1 term would not be needed.

Now, we can upper bound the expected number of tests as follows

$$
\begin{aligned}
\mathbb{E}[l(\omega)] &= \sum_{\omega \in \Omega} p(\omega) l(\omega) \\
&\leq \frac{H(Y) - |\log_2 \gamma| + \left( \sum_\omega \mathbb{P}(\omega) m(\omega) \right) \log_2 \frac{1-\alpha}{\alpha}}{\log_2(1-\alpha) - \log_2(\frac{1}{2} + \delta)} + 1 \\
&\leq \frac{H(Y) - |\log_2 \gamma| + \alpha|\mathcal{T}| \log_2 \frac{1-\alpha}{\alpha}}{\log_2(1-\alpha) - \log_2(\frac{1}{2} + \delta)} + 1.
\end{aligned}
\tag{33}
$$

We used equation 32 to obtain the upper bound in the first inequality. The second inequality is obtained by observing that $m(\omega)$ can be point-wise upper bounded by another random variable which denotes the number of test outcomes that would disagree with their corresponding true outcomes, if all the tests in $\mathcal{T}$ were carried out. As a result, we can upper bound the the expectation of $m(\omega)$ by $\alpha|\mathcal{T}|$, which is the mean of a binomial random variable with $|\mathcal{T}|$ trials each having success probability $\alpha$. □

### A.4 More details about the Dasgupta Bound in equation 7

Observe that

$$
H(Y) \leq \text{opt}(\mathcal{T}, Y); \quad \ln |\mathcal{Y}| \leq \ln \left( \frac{1}{\min_Y p(Y)} \right),
\tag{34}
$$

where the first inequality is due to the source coding theorem (Shannon, 1948) and the second equality is obtained by observing that $\min_Y p(Y)$ is always upper bounded by $\frac{1}{|\mathcal{Y}|}$ since the probabilities have to sum to 1, where $|\mathcal{Y}|$ is the number of distinct states of $Y$. Thus, combining equation 34 and equation 7 we get the following bound,

$$
\text{B}_{\text{Das}} \geq 4H(Y) \ln |\mathcal{Y}|,
\tag{35}
$$

Comparing equation 35 with our bound in equation 5 we conclude that Dasgupta's bound would be looser (larger) that our bound for

$$
\delta \leq 2^{-\frac{1}{4 \ln |\mathcal{Y}|}} - \frac{1}{2}.
\tag{36}
$$

### A.5 Experiments

In this section we evaluate the tightness of our bound for the case when tests are functions of $Y$, along with those reported previously in literature, on some practical instantiations of the information maximization algorithm. Both examples are inspired from the classical "twenty questions" (20Q) game where one player thinks of an entity, and the goal of the other player is to guess the object correctly by asking the minimum number of questions about the object.[5]

#### A.5.1 20Q with birds

In our first example, we play 20Q with birds. One player thinks of a bird species $Y$ and the other player asks questions about different visual attributes about the chosen bird in order to identify $Y$. For this purpose we use the CUB-2011 dataset (Wah et al., 2011). The dataset consists of images of 200 different bird species, each annotated with answers to 312 binary questions about visual attributes like belly colour, wingspan, beak shape etc. It is reasonable to assume that given all 312 attributes, $Y$ is determined and that every visual attribute question is a function of $Y$. However, the image annotations are noisy. To remedy this, in accordance with prior work (Koh et al., 2020), we modify the annotations in the following manner; if more than half the images for a particular class has value $x$ for a certain attribute, we set the annotation for that attribute of all images from that class to $x$.

Given the problem setup, each test corresponds to a binary visual attribute question, and the outcome of the test is the corresponding answer. We carry out information maximization to sequentially

---
[5]The tests in this context are the questions and their respective outcomes are the answers to the question.

(a)  $Y$ = "Blue Jay"

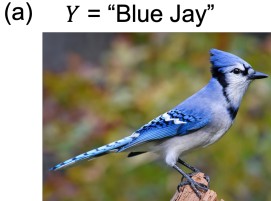

(b)  $Y$ = "leopard"

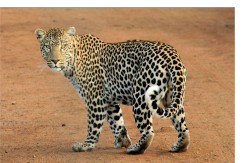

1. Is back pattern solid? Yes
2. Is under tail colour black? No
3. Is breast colour white? No
4. Is bill colour black? Yes
5. Is belly colour yellow? No
6. Is belly colour grey? No
7. Is breast colour black? Yes
8. Is wing colour blue? Yes

1. Is it lean? Yes
2. Is it small? No
3. Is it white? No
4. Is it yellow? Yes
5. Does it have patches? Yes
6. Is it black? Yes

Figure 6: Example runs of the greedy strategy on the two tasks considered here; **(a)** 20Q with birds; **(b)** 20Q with animals. One player thinks of a label $Y$, and the other player carries out tests by asking questions about $Y$ in order to correctly identify it. The test outcomes are obtained from ground truth annotations.

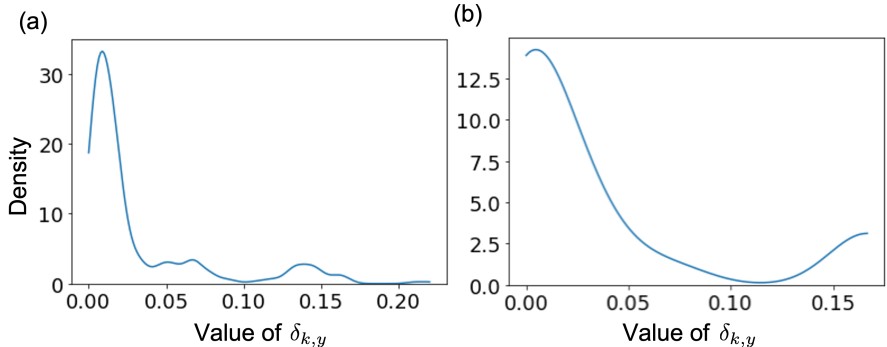

Figure 7: Distribution for $\delta_{k,y}$ over all iterations and labels $y \in \mathcal{Y}$ for the two examples (tasks) considered here; **(a)** 20Q with birds; **(b)** 20Q with animals. This same data was used in our plots in Figure 1 in the main paper.

carry out tests until the species, $Y$, has been determined. We use the empirical probabilities in the dataset to compute all the entropic quantities required for running the greedy strategy (Algorithm in equation 3). Figure 6a shows an example run of the greedy strategy for this task.

For every species $y \in \mathcal{Y}$, for every iteration $k$ and selected test $T_{k,y}$, we record the quantity $\delta_{k,y} := \left| P(T_{k,y} = 1 \mid t_{1:k-1}) - \frac{1}{2} \right|$. Figure 7a shows the distribution for $\delta_{k,y}$ over all iterations and labels $y \in \mathcal{Y}$. It is clear from the figure that the given set of tests for this example is $\delta$-unpredictable for $\delta = 0.22$, since for this value all tests selected by the greedy strategy (in any iteration for any data-point) will satisfy equation 2. We compare the mean number of queries needed by the greedy strategy for this task with various upper bounds to evaluate their tightness in Table 1.

We make the following observations.

1. As discussed in §4.2, $B_{Das}$ is a vacuous upper bound (161.86) compared to the true value of the average number of tests needed to identify $Y$ using the information maximization algorithm (7.70). This is because it makes no assumptions about $\mathcal{T}$, whereas it is clear from Figure 7a that the tests selected satisfy equation 2 for $\delta = 0.22$.

Table 1: Comparison of the average number of tests needed by the greedy strategy to identify $Y$, for the two tasks considered here, with those predicted from upper bounds. We also report the entropy of $Y$ for each task (column 2) which is a lower bound on the best achievable performance. InfoMax (column 2) refers to the observed empirical performance of the information maximization algorithm (the greedy strategy) in terms of the average number of tests needed to identify $Y$. For $B_{Das}$ we evaluate the lower bound in equation 35 since computing the optimal strategy is intractable.

| Task | $\delta$ | $H(Y)$ | InfoMax | $B_{Ours}$ | $B_{Lov}$ | $B_{Das}$ |
|---|---|---|---|---|---|---|
| 20Q with birds | 0.22 | 7.64 | 7.70 | **16.12** | 17.44 | 161.86 |
| 20Q with animals | 0.17 | 5.64 | 5.73 | **9.66** | 12.69 | 88.31 |

2. For evaluating both $B_{Lov}$ and $B_{Ours}$ we use $\delta = 0.22$, since this value makes the set of tests considered here $\delta$-unpredictable (see Figure 7a). Though, our bound is tighter 16.12, $B_{Lov}$ obtains a similar value of 17.44. This is because as we saw in Figure 2, at about $\delta \approx 0.2$ and large $|\mathcal{Y}|$ (In this example $|\mathcal{Y}| = 200$), both the bounds are roughly the same.

### A.5.2 20Q WITH ANIMALS

For our second example, we play 20Q with animals. For this purpose, we use the AwA2 dataset (Xian et al., 2018). The datasets consists of images of 50 different animal classes each annotated with answers to 85 binary attributes such as number of legs, skin color, eating habits, habitat etc. Every attribute answer is a deterministic outcome of the label $Y$ and together they determine $Y$, that is, knowing the answers to all 85 attribute questions allows for identifying $Y$.

Given this description, we take every binary attribute question in the dataset to be test, and its answer to be the corresponding outcome of the test. We follow the same steps as before, namely, carry out the greedy strategy by using the empirical probabilities for computing the relevant entropy terms and record $\delta_{t,k}$ for every iteration $k$ and for every label $y \in \mathcal{Y}$. Figure 7b shows this distribution. Accordingly we pick $\epsilon = 0.167$ for evaluating $B_{Lov}$ and $B_{Ours}$. Figure 6b shows an example run of the greedy strategy for this task.

The comparisons of different bounds are reported in Table 1. We observe the same trend as the previous task. $B_{Das}$ gives a vacuous bound compared to the true value (column 2, InfoMax). Our bound $B_{Ours}$ is tighter than $B_{Lov}$ even though both are evaluated at the same $\delta = 0.167$.

