# OpenReview forum: "Performance Bounds for Active Binary Testing with Information Maximization"
_ICLR.cc/2024/Conference — Submitted to ICLR 2024_

### Official Review · Reviewer_fURR · 2023-10-19

**Soundness:** 2 fair
**Presentation:** 3 good
**Contribution:** 2 fair
**Rating:** 5
**Confidence:** 4

**Summary:**

This paper aims to provide a tighter bound for the well-known active binary testing with information maximization (Informax). The approach is similar to Garey and Graham (1974) or Loveland (1985) which uses a key assumption that the sequence of tests are $(\gamma,\delta)$-unpredictable to derive the minimum expected number of binary tests to predict a target variable $Y$.

**Strengths:**

+ For oracle binary tests, the proposed bound can improve the existing bounds for the same setting such as Garey and Graham (1974) or Loveland (1985). The most interesting contribution is to reduce $\log_2(|\mathcal{Y}|)$ to $H(Y)$ (cf. Theorem 1).
+ Experiments on datasets CUB-2011 (20Q with birds) and AwA2 (20Q with animals) are provided, which demonstrate that the proposed bounds (for oracle tests) can be better than Garey and Graham (1974) or Loveland (1985)'s counterparts.
+ The authors give a new bound for the noisy binary tests (cf. Theorem 4), and there haven't any existed bounds for this model.

**Weaknesses:**

+ It is hard to think of how to design a binary test sequence which is $(\gamma,\delta)$-unpredictable although this is the key assumption to achieve results in this paper. Hence, the proposed bounds (for both oracle and noisy tests) do not guide us how to design an active binary test sequence based on Informax principle to achieve them.
+ The tightness of the given bound also depends on $\delta$. However, in general, it looks hard to find the optimal value of $\delta$ for an existing sequence of binary tests.
+  In the two provided experiments, the authors assume that $Y$ is uniformly distributed on some finite set $\mathcal{Y}$, so $H(Y)=\log_2(|\mathcal{Y}|)$. Therefore, the improvements of the authors' bound in Theorem 1 over Loveland's bound (cf. 6), which are shown in Fig. 2 (or Table 1  in Appendix), is mainly originated from a better control of constant factor (depending on $\delta$). The main interesting contribution that reduces $\log_2(|\mathcal{Y}|)$ to $H(Y)$ is not shown in these experiments.
+ In Section 5, the authors mention some obtained bounds for noisy tests (via BSC), which are achieved based on the decomposition $T(Y)=D_T(Y)\oplus N_T$. The design of binary tests to achieve these bounds is based on having knowledge of $D_T(Y)$ (Lemma 2) or $I(T;Y|\mathcal{A}_{1:t})$ (cf. (1)), which looks very hard to obtain in practice. In addition, the tightness of these bounds is not verified in the paper.

**Questions:**

I don't have any question. Please see the weaknesses above and let me know if I misunderstand anything.

Some typos and improvements:

+  Repetition in (27) and (28). Please remove the redundancy.
+  You should mention that $T(Y) \in \{0,1\}$ for any test $T$ in Section 3. This means that your results are limited to the binary test (yes/no questions).

---

> ### Author Response · Authors · 2023-11-23
> **Response to Reviewer fURR**
>
> Thank you for your time and feedback. We will address each of your comments (which are highlighted in bold) below:
>
> **The main interesting contribution that reduces $\log_2(|\mathcal{Y}|)$ to $H(Y)$ is not shown in these experiments.**
>
> Thank you for this suggestion to help improve the presentation of our results. We have added Figure 3 in the updated paper which supports our claim that as $Y$ becomes more non-uniform (entropy reduces), our bound is much tighter than Loveland's previous bound.
>
> **The $\delta$-unpredictable property does not guide how to design set of tests that satisfy this property for both the oracle and the noisy case.**
>
> We agree and this has been discussed as a limitation of our current work. As discussed in the Introduction, without any asumptions on the set of tests $\mathcal{T}$, it has been shown in prior work (Loveland, 1985) that the performance of the greedy strategy can be arbitrarily bad compared to the optimal strategy. However, this is often not what's observed in practice. In this work we identify $\delta$-unpredictability to be a key property that practically employed set of tests used by the information maximization algorithm often exhibit (Geman et al., 2015). We then prove that assuming this property holds, the performance of the greedy strategy (in terms of the avg. number of tests needed for prediction) comes within a constant factor of entropy in the oracle case and within a constant factor of entropy plus a term that depends on the noise in the noisy case. We believe these results are nearly-optimal since entropy is a lower bound on the performance of any strategy for this problem and hope sheds some light on why the greedy strategy works so well in practice. Future work would try to develop testable conditions that could verify if a given $\mathcal{T}$ is $\delta$-unpredictable or not.
>
> **Repetition in (27) and (28). Please remove the redundancy.**
>
> Thank you for noticing this. This redundancy has been removed.
>
> **You should mention that for any test $T$, $T(Y) \in \{0,1\}$ in Section 3. This means that your results are limited to the binary test (yes/no questions).**
>
> This has already been mentioned 2 lines after equation 1 in Section 3 - $t_{k+1} \in \{0,1\}$.

---

### Official Review · Reviewer_NgKX · 2023-11-01

**Soundness:** 3 good
**Presentation:** 4 excellent
**Contribution:** 3 good
**Rating:** 6
**Confidence:** 2

**Summary:**

This paper studies the problem of predicting a random variable using tests. Specifically, the authors analyse the commonly used greedy heuristic of information maximization under the assumption that the set of tests are $\delta$-unpredictable. The main contribution of the paper is new upper bounds on the number of tests needed for information maximazation under both the oracle tests and the noisy tests. The obtained bound for oracle tests is tighter in certain regime of parameters than previous bounds, while the bound for noisy tests is the first such results.

**Strengths:**

1. The paper is very well-written. I really appreciate the authors for including proof sketch and discussion of high-level ideas, which makes the paper easy-to-follow even for readers that are not familiar of the problem of active testing.
2. I think understanding the performance of greedy heuristic that has practical application is an important question. The obtained bound for oracle tests gives tighter guarantee than previous results in certain regimes of parameters and the paper presents detailed comparison with previous bounds. This paper is also the first to obtain bound for information maximization for noisy tests.

**Weaknesses:**

1. As the authors comment the limitation section, the assumption that the tests are $\delta$-unpredictable is not very useful in practice since it is not know how to compute the corresponding $\delta$.
2. The assumption of i.i.d. noise for noisy tests also limits the practical application of the results since the noise is often dependent on the value of $Y$ and the tests outcomes are not independent.

**Questions:**

Does the authors have any insights in resolving weaknesses mentioned above?

---

> ### Author Response · Authors · 2023-11-23
> **Response to Reviewer NgKX**
>
> Thank you for your time and feedback. We will address each of your comments (which are highlighted in bold) below:
>
> **As the authors comment the limitation section, the assumption that the tests are \delta-unpredictable is not very useful in practice since it is not know how to compute the corresponding $\delta$.**
>
> We believe that there needs to be some assumption on the structure of the tests to be able to compute the corresponding $\delta$. One such structure that we believe might be useful is a hierarchical structure between the tests. Specifically, say the tests are arranged in a hierarchy such that a test is true only if all its ancestor tests are true. Similarly, if a test is false, then all tests in the sub-hierarchy rooted at this test will be false. With such a structure, one can obtain several interesting relationships between the tests. For instance, one can immediately conclude that the conditioned on a set of test outcomes, the probability a test is true is always greater than the probability any of its children tests are true. While more work needs to be done to obtain precise relationship between the hierarchy and $\delta$, we believe this to be a potential direction to investigate in resolving this weakness.
>
> **The assumption of i.i.d. noise for noisy tests also limits the practical application of the results since the noise is often dependent on the value of $Y$ and the tests outcomes are not independent.**
>
> We agree with the reviewer, our noise model is meant to be a first step at analyzing the performance of the greedy strategy under noise. The major challenge with using more complicated noise models that depend on $Y$ is that the mutual information term becomes analytically intractable. We believe variational approximations developed to estimate mutual information can offer tractable alternatives to analyze the greedy strategies' performance in such settings.

---

### Official Review · Reviewer_6HBL · 2023-11-01

**Soundness:** 3 good
**Presentation:** 3 good
**Contribution:** 2 fair
**Rating:** 5
**Confidence:** 4

**Summary:**

The paper considers the problem of identifying the value of a random variable $Y$ through a sequence of binary tests. The paper focuses on studying the Information Maximization procedure where at each step we greedily choose the test that maximizes the conditional mutual information.

The paper first considers the case where the sets of allowed tests are deterministic functions (called oracle tests), and then studies the case where the output of the oracles are corrupted by noise. In both cases, the paper assumes that the set of tests satisfies a "$\delta$-unpredictability property" where at each $k$-th stage of the InfoMax procedure, it is always possible to find a test $T_k$ such that $Pr[T_k=1|T_1,\ldots,T_{k-1}]$ is at most $\delta$ away from 1/2 (unless of course we already identified $Y$ at the desired accuracy). In other words, we can always find a test that approximately bisects the set of possible values of $Y$.

Assuming that the set of tests satisfies the $\delta$-unpredictability property, the paper proves an upper bound on the expected number of tests needed to identify $Y$. The bound depends on $\delta$ and is proportional to the entropy $H(Y)$. In the noisy case, the bound also depends on the noise-level $\alpha$.

**Strengths:**

The problem considered in the paper is interesting, and the bound that is given is optimal up to constant factors since it is proportional to the entropy H(Y).

**Weaknesses:**

I have to admit that I am not very familiar with the literature of this topic in particular, but from an information-theoretic perspective, the novelty/contribution is a bit limited: The techniques used in the paper are very simple and the results are not too surprising.

**Questions:**

Did the authors consider extending the work to more general tests where a test consists of passing $Y$ through a noisy channel of input alphabet $\mathcal{Y}$ (the set of possible value of $Y$) and of output alphabet $\{0,1\}$?

---

> ### Author Response · Authors · 2023-11-23
> **Response to Reviewer 6HBL**
>
> Thank you for your time and feedback. We will address each of your comments (which are highlighted in bold) below:
>
> **I have to admit that I am not very familiar with the literature of this topic in particular, but from an information-theoretic perspective, the novelty/contribution is a bit limited**
>
> We are not sure why the reviewer feels the novelty of our work is limited from an information-theoretic perspective. We provide a bound on the performance of the greedy strategy that is within a constant factor of the entropy of $Y$, H(Y), which has not been established a priori in literature. Moreover, we show that our bound improves upon previous known bounds in literature. We provide more context for our work in the next paragraph.
>
> It is well-known that when one has access to all possible binary functions of $Y$ as tests, the greedy strategy needs at most $H(Y) + 1$ tests to identify $Y$. However, in practical scenarios one never has access to all possible binary functions of $Y$. Take the example of disease diagnosis. $Y$ refers to the disease of the patient and the tests are about different symptoms the patient might be experiencing. The set of all binary functions of $Y$ (power set of the set of all diseases) is a much larger set than the set of possible symptoms a patient might experience from. In such restricted settings, there is no guarantee on the performance of the greedy strategy. This work is an effort to gain some theoretical understanding in such settings by imposing assumptions on the distribution of the set of tests and $Y$. The assumptions used in this paper are informed from the observation that often in practical scenarios the set of tests obtained is $\delta$-unpredictable for modest values of $\delta$, say between 0.15-0.3.
>
> **Did the authors consider extending the work to more general tests where a test consists of passing $Y$ through a noisy channel of input alphabet $\mathcal{Y}$ (the set of possible value of $Y$) and of binary alphabet 0,1**
>
> We hope that the results presented in our paper motivate future research into this direction where more general noise models are considered. This is discussed in the conclusion & limitations section of the paper. The main challenge with more complicated noise models beyond the i.i.d. assumption in this paper is that then the mutual information terms are no longer analytically tractable which complicates analysis.

---

> > ### Comment · Reviewer_6HBL · 2023-12-02
> >
> > Regarding the novelty of the work: I am not questioning that the result itself is new. My comment was mainly about the information-theoretic techniques that were used/developed and how sophisticated/advanced they are. From that regard, I find the proofs to be a bit simple and in that sense one would say that the information-theoretic contributions are a bit limited. This does not mean that the paper does not have other important contributions.
> >
> > Having simple proofs is not necessarily a bad thing and in some contexts it can be a very good thing, for example:
> >
> > If the problem has been extensively considered in the past and the research community had overlooked the simple proof.
> > If the work introduces a new problem that is very interesting to pursue in its own right, and the introduction of the work introduces many open research questions which would be of interest for the community.
> > However, I am not sure if these cases do apply to the paper and I cannot properly assess the other contribution aspects of the paper (other than the the sophistication of the proofs) because I am not very familiar with the literature of this particular topic that is studied, and this is why I added the disclaimer "I have to admit that I am not very familiar with ...". I invite the chair to take this into account. I only provided my review as an information theorist.

---

### Official Review · Reviewer_xWDX · 2023-11-03

**Soundness:** 3 good
**Presentation:** 2 fair
**Contribution:** 3 good
**Rating:** 5
**Confidence:** 3

**Summary:**

This paper deals with the problem of determining the value of a random variable Y, by adaptively performing a series of tests with binary output. The goal is to minimize the expected number of tests needed to determine the value of Y with the desired confidence. In the case where any possible binary test on the values of $Y$ is available, this problem has been shown to be almost optimally solvable with at most $H(Y)+1$ tests, where $H(Y)$ denoted the entropy, via the information maximization strategy (i.e each time selecting the test with probability closest to $1/2$). However, the authors consider the more practical setting where a specific set $\mathcal{T}$ of tests is available. The results of the paper identify sufficient conditions that this family of tests has to satisfy in combination with the expected number of tests needed. In particular, the notion of $\delta$-unpredictability is considered for the test families, where $\delta$ is a measure of uncertainty for the test outcomes. The main result is that one can identify the value of Y using in expectation $\frac{H(Y)}{\log (½+\delta)^{-1}}$ test from a $\delta$-unpredictable family using the greedy information maximization strategy. This setting is also extended to the case where the test outcomes are noisy as a result of independently passing through a binary symmetric channel and a similar result is shown involving an additional parameter $\gamma$ representing the target confidence for the value of $Y$.

**Strengths:**

The paper deals with a fundamental problem from the perspective of more realistic and practical settings than the ones previously considered including a noise model.

**Weaknesses:**

There is no discussion about lower bounds on the number of tests needed for either the noisy or the oracle (noiseless) case. I believe one should be able to derive something using information theory, but it's not clear to me if those bounds would match the upper bounds in the paper.
The presentation could be improved since the results and contributions are not entirely clear form the introduction.


Minor cpmments
-In Theorem 1 (and similarly for Theorem 4): The use of absolute value in the denominator is confusing since the expressing inside is always negative.    I suggest using $\log (½+\delta)^{-1}$ instead.
-Page 3, "noisy tests" paragraph, line 7: By "pre-noise" did you want to say "de-noise"?
-Page 4, line 17: the word "after" is probably missing after the word "or"

**Questions:**

1. Are there any results for the case where all tests are chosen (non adaptively) in the beginning?
2. Can the expression in Theorem 1 (and similarly Theorem 4) be written with respect to the entropy $H(½+\delta)$ of a Bernoulli distribution? One would expect this because this seems to be the amount of information revealed with each test.

---

> ### Author Response · Authors · 2023-11-23
> **Response to Reviewer xWDX (1/2)**
>
> Thank you for your time and feedback. We will address each of your comments (which are highlighted in bold) below:
>
> **There is no discussion about lower bounds on the number of tests needed for either the noisy or the oracle (noiseless) case. I believe one should be able to derive something using information theory, but it's not clear to me if those bounds would match the upper bounds in the paper.**
>
> Given a random variable $Y$, a lower bound on the average number of tests needed (regardless of the set of tests available) is given by the entropy of $Y$, $H(Y)$. This means that no strategy (greedy or otherwise) can do better than this. This is mentioned in the second paragraph of the Introduction.
>
> Our contribution is to show that when one has access to a set of oracle tests that satisfy a property called $\delta$-unpredictability, then the average number of tests needed by the information maximization strategy is within a constant factor of $H(Y)$, where the constant depends logarithmically on $\delta$. We believe this upper bound to be nearly optimal given the entropic lower bound mentioned in the previous paragraph.
>
> Finally, lower bounds specific to a given set of tests that are $\delta$-unpredictable (for both the noisy and the oracle case) is an interesting direction for future research. The main difficulty in showing such lower bounds lies in the difficulty in constructing a set of tests that is $\delta$-unpredictable for a given value of $\delta \in [0, 0.5]$. This has been discussed as a limitation of our current work.
>
> **The presentation could be improved since the results and contributions are not entirely clear form the introduction.**
>
> We have rewritten our contributions in the Introduction section to make this more clear (annotated in blue). Our main contributions are as follows:
>
> 1. We first study the oracle case where tests are functions of $Y$. Assuming the given set of tests, $\mathcal{T}$, is $\delta$-unpredictable for some $\delta \in [0, \frac{1}{2}]$, we prove that the greedy strategy needs at most $\frac{H(Y)}{-\log_2(\frac{1}{2} + \delta)}$ number of tests on average to identify (predict) $Y$. To the best of our knowledge, this is a first bound on the performance of the greedy strategy that explicitly depends on the entropy of $Y$. This is desirable since a lower bound on the average number of tests needed for any given $\mathcal{T}$ is given by the entropy of $Y$ (Shannon, 1948). Moreover, we show that our bound is tighter than previously known bounds for oracle tests in practically relevant settings.
>
> 2. We then extend our analysis to the noisy case where we assume that test outcomes are corrupted via a binary symmetric channel. We obtain an upper bound on the performance of the greedy strategy that explicitly depends on $\delta$ and the noise level. Specifically, our bound in this case is again within a constant factor of the entropy of $Y$ modulo an additional term, where the constant factor and the additional term depend on $\delta$ and the noise level. To the best of our knowledge, this is the first such result for the greedy strategy given noisy tests.
>
> **Minor comments -In Theorem 1 (and similarly for Theorem 4): The use of absolute value in the denominator is confusing since the expressing inside is always negative. I suggest using $\log_2(\frac{1}{2} + \delta)^{-1}$ instead. -Page 3, "noisy tests" paragraph, line 7: By "pre-noise" did you want to say "de-noise"? -Page 4, line 17: the word "after" is probably missing after the word "or"**.
>
> Thank you for these suggestions and we have implemented these changes in our updated paper.
>
> **Can the expression in Theorem 1 (and similarly Theorem 4) be written with respect to the entropy of a Bernoulli distribution? One would expect this because this seems to be the amount of information revealed with each test.**
>
> For Theorem 1 yes! In fact, a reviewer of an earlier version of this paper provided us with such a result which is reported in Theorem 6 (along with a proof) in the appendix. More specifically, the result says that the number of tests needed in the oracle case is upper bounded by $\frac{H(Y)}{h(\frac{1}{2} + \delta)}$, where $h(\frac{1}{2} + \delta)$ is the entropy of a Bernoulli random variable. Since this was not our result we felt it would not be ethical to include it in the main paper.
>
> Finally, it is not straightforward to extend that result to Theorem 4. This is because in the presence of noise, Eq. 14 (in the proof) is no longer valid, and requires further assumptions on the joint distribution of the test outcomes and $Y$ than the $\gamma$ stopping criterion assumed in this paper.

---

> ### Author Response · Authors · 2023-11-23
> **Response to Reviewer xWDX (1/2)**
>
> **Are there any results for the case where all tests are chosen (non adaptively) in the beginning?**
>
> Yes, it is known that if the test outcomes are conditionally independent given $Y$, then a greedy selection based on mutual information (this is done in expectation without observing any outcomes unlike the adaptive algorithm studied in this paper) is known to produce a $1 - 1/e$ approximation to the optimal solution (Krause and Guestrin (2005)). This result is obtained by using the theory of submodular functions since the mutual information (in expectation over test outcomes) when test outcomes are conditionally independent given $Y$ is a sub-modular function. Interestingly, mutual information when conditioned on actual test outcomes (not in expectation) is no longer submodular (Chen et al., COLT 2015), which is the case in adaptive strategies.
>
> However, compared to fixed strategies, adaptive strategies like the information maximization algorithm are empirically found to be more efficient since subsequent choices of tests are now conditioned on outcomes observed so far. Nevertheless, this empirical experience is poorly understood in theory and we hope the results presented in our paper help shed some light on this matter.

---

### Author Response · Authors · 2023-11-23
**Changes in the paper post review**

We thank all reviewers for their time, effort and valuable feedback that helped improve the presentation of this work. We are glad reviewers found our work very well-written (Reviewer NgKX), results novel and interesting (Reviewer fURR & NgKX). In light of the reviews, we have made the following changes to the paper (annotated in blue).

1. Rewrote our contributions in the introduction to make them more clear as requested by Reviewer xWDX.
2. Added an experiment to verify the improvement of our bounds over prior work in the scenario the distribution over $Y$ (the labels) is not uniform as requested by Reviewer fURR.

---

### Meta-Review · Area_Chair_Rn5J · 2023-12-10

**Metareview:**

This paper addresses the problem of predicting a binary variable via an information perspective. Despite the interesting setting and the information-theoretic approach, the paper can be further improved with better organisation and a clearer explanation on the theoretical bounds. The reviewer's concerns are not fully resolved; and the paper may not be ready to appear in the conference in its current form.

**Justification For Why Not Higher Score:**

The organisation of the paper can be improved and the contribution and significance can be explained more clearly.

**Justification For Why Not Lower Score:**

N/A

---

### Decision · Program_Chairs · 2024-01-16

Reject